The first gladius-bearing coleoid cephalopods from the lower Toarcian “Schistes Cartons” Formation of the Causses Basin (southeastern France)

Jattiot Romain romain.jattiot@u-bourgogne.fr 1 2
Coquel-Poussy Nathalie 3
Kruta Isabelle 1
Rouget Isabelle 1
Rowe Alison J. 1
Moreau Jean-David 2
1 Centre de Recherche en Paléontologie –Paris (CR2P), MNHN, CNRS, Sorbonne Université , Paris , France
2 Biogéosciences, CNRS, Université Bourgogne , Dijon , France
3 Saint Bauzile , France
Żyła Dagmara
Electronic publication date: 2024 Feb 26
Publication date: 2024
Volume: 12
Electronic Location ID: e16894
Received 2023 Nov 24; Accepted 2024 Jan 16
Copyright: ©2024 Jattiot et al.
Copyright year: 2024
Copyright holder: Jattiot et al.
License: This is an open access article distributed under the terms of the Creative Commons Attribution License, which permits unrestricted use, distribution, reproduction and adaptation in any medium and for any purpose provided that it is properly attributed. For attribution, the original author(s), title, publication source (PeerJ) and either DOI or URL of the article must be cited.
License URL: https://creativecommons.org/licenses/by/4.0/

Keywords: Coleoids, Lower Toarcian, Causses Basin, Paleobiogeography, Schistes Cartons

Funding: The authors received no funding for this work.

==============================
The fossil record of gladius-bearing coleoids is scarce and based only on a few localities with geological horizons particularly favourable to their preservation (the so-called Konservat-Lagerstätten), which naturally leads to strongly limited data on geographical distributions. This emphasizes the importance of every new locality providing gladius-bearing coleoids. Here, we assess for the first time the gladius-bearing coleoid taxonomic diversity within the lower Toarcian “Schistes Cartons” of the Causses Basin (southeastern France). The material includes two fragmentary gladii, identified as Paraplesioteuthis sagittata and ?Loligosepia sp. indet. Just with these two specimens, two (Prototeuthina and Loligosepiina) of the three (Prototeuthina, Loligosepiina and Teudopseina) suborders of Mesozoic gladius-bearing coleoids are represented. Thus, our results hint at a rich early Toarcian gladius-bearing coleoid diversity in the Causses Basin and point out the need for further field investigations in the lower Toarcian “Schistes Cartons” in this area. This new record of Paraplesioteuthis sagittata is only the second one in Europe and the third in the world (western Canada, Germany and now France). Based on these occurrences, we tentatively suggest that P. sagittata originated in the Mediterranean domain and moved to the Arctic realm through the Viking Corridor to eventually move even farther to North America.

Introduction

Within the cephalopod subclass Coleoidea Bather, 1888, the cohort Neocoleoidea Haas, 1997 includes present-day organisms (e.g., vampire squid, octopods, squids, cuttlefishes and their relatives) that are mainly characterized by their internal shell. The latter is called gladius and consists of a sturdy but flexible chitinous structure within the dorsal mantle. The evolutionary history, anatomy and paleobiology of Mesozoic gladius-bearing coleoids has been extensively studied in the last two decades (Fischer & Riou, 2002; Haas, 2002; Košťák, 2002; Fuchs, Keupp & Engeser, 2003; Fuchs, Engeser & Keupp, 2007a; Fuchs, Klinghammer & Keupp, 2007b; Fuchs et al., 2007c; Fuchs, 2009; Fuchs & Iba, 2015; Fuchs et al., 2016; Bizikov, 2004; Bizikov, 2008; Fuchs & Weis, 2004; Fuchs & Weis, 2008; Fuchs & Weis, 2009; Fuchs & Weis, 2010; Wilby et al., 2004; Riccardi, 2005; Fuchs, 2006a; Fuchs, 2006b; Fuchs, 2006c; Fuchs, 2007; Fuchs, 2009; Fuchs, 2015; Fuchs, 2016; Fuchs, 2019; Fuchs, 2020; Fuchs & Schultze, 2008; Larson, 2010; Donovan & Strugnell, 2010; Keupp et al., 2010; Klug, Schweigert & Dietl, 2010; Klug et al., 2015; Klug et al., 2021a; Klug et al., 2021b; Fuchs & Larson, 2011a; Fuchs & Larson, 2011b; Schlögl, Košťák & Hyžny, 2012; Breton, Strugnell & Donovan, 2013; Donovan & Boletzky, 2014; Fuchs & Iba, 2015; Jattiot et al., 2015a; Donovan & Fuchs, 2016; Kruta et al., 2016; Marroquín, Martindale & Fuchs, 2018; Košťák et al., 2021; Moreau et al., 2022; Rowe et al., 2022; Rowe et al., 2023) thanks to a few localities with geological horizons particularly favourable to their preservation, the so-called Konservat-Lagerstätten. Thus, studies on exceptionally preserved Mesozoic gladius-bearing coleoids are based on Konservat-Lagerstätten such as the Lower Jurassic Posidonia Shales of Holzmaden, Germany (e.g., Hauff & Hauff, 1981; Riegraf, Werner & Lörcher, 1984; Klug et al., 2021a), the Middle Jurassic of Christian Malford and Rixon Gate, England (e.g., Wilby et al., 2004), the Middle Jurassic of La Voulte-sur-Rhône, France (e.g., Fischer & Riou, 1982; Fischer & Riou, 2002; Charbonnier, 2009; Kruta et al., 2016; Rowe et al., 2022; Rowe et al., 2023), the Upper Jurassic of Eichstätt, Solnhofen, Painten and Nusplingen Plattenkalks, Germany (e.g., Fuchs, 2006b; Fuchs, 2015; Klug, Schweigert & Dietl, 2010; Klug et al., 2015; Keupp et al., 2010), the Upper Jurassic of the Causse Méjean, France (Moreau et al., 2022) and the Upper Cretaceous of Hâkel, Hâdjoula and Sâhel Aalma, Lebanon (e.g., Fuchs, 2006c; Fuchs, Bracchi & Weis, 2009; Fuchs & Larson, 2011a; Fuchs & Larson, 2011b; Jattiot et al., 2015a; Klug et al., 2021b).

However, as these Lagerstätten are not continuous in time and space, the record of Mesozoic gladius-bearing coleoids is intermittent. As noted by Fuchs et al. (2016), our knowledge on geographical distributions of species still needs to be greatly improved, which confers great importance upon every new locality yielding gladius-bearing coleoids.

Here, we describe fossils of gladius-bearing coleoids from the lower Toarcian “Schistes Cartons” Formation (contemporaneous with the famous Posidonia Shales of Holzmaden; see references above) of the Causses Basin in southeastern France. The material described here represents the first gladius-bearing coleoid remains documented from this area. Thus, this work is the first attempt to assess the taxonomic diversity of gladius-bearing coleoids from the lower Toarcian “Schistes Cartons” of the Causses Basin. These findings also allow to provide new insights on the paleobiogeography of Toarcian gladius-bearing coleoids and are a first step towards discussions on hypothetical migration corridors.

Materials & Methods

The material studied herein was collected by one of us (N.C.P.) in the northern part of the Causses Basin, Lozère, France (more precisely in the vicinity of Saint Bauzile village in the Valdonnez valley, at the base of the Balduc plateau, 5 km southern of Mende; Fig. 1). It consists of two gladius-bearing coleoid specimens (M486_2024.1.1 and M486_2024.1.2) preserved as compressions on slabs, which are housed in the paleontological collections of the Musée du Gévaudan (Mende, Lozère). We used UV-light to highlight soft tissues, using a UV wavelength of 360 nm. The anatomical terminology of the gladius and systematic paleontology follows Fuchs & Weis (2010), Fuchs & Larson (2011a), Fuchs & Larson (2011b), Fuchs (2016) and Fuchs (2020). Measured characters (see Appendix S1A) are: preserved gladius length, maximum gladius width, median field widthhypz, hyperbolar zone length, maximum lateral fields width. These measurements are standard in most published studies on gladius-bearing coleoids (e.g., Fuchs & Larson, 2011a; Fuchs & Larson, 2011b; Fuchs, 2016; Fuchs, 2020).

Figure 1 Geographical location of the Causses Basin.

The black star indicates the Saint Bauzile site, where the studied material was retrieved. Image source credit: Moreau et al. (2021) ©Cambridge University Press, reproduced with permission.

Geological settings and stratigraphy

The Causses Basin area is constituted by Jurassic limestone plateaus that are located south of the Massif Central (southeastern France). The studied material was collected in situ near Saint Bauzile (Fig. 1) from a ravine exhibiting a 27 m thick stratigraphic section showing upper Pliensbachian to middle Toarcian deposits (Lower Jurassic, Fig. 2). The upper Pliensbachian corresponds with the Villeneuve Formation and consists of grey marls alternating with nodular and lenticular limestone beds (Fig. 2). At Saint Bauzile, this formation mainly yields large belemnite rostra, bivalves (Plicatula (Harpax) spinosa (Sowerby, 1812–1822)), brachiopods (e.g., Cirpa boscensis (Reynès, 1868)) as well as some ammonites (e.g., Juraphyllites, Pleuroceras) and invertebrate burrows (Tisoa siphonalis de Serres, 1840). The Toarcian is divided into two formations, the “Schistes Cartons” Formation (lower Toarcian) and the Marnes de Fontaneilles Formation (lower to upper Toarcian). The “Schistes Cartons” Formation is about 7.5 m thick and consists of dark grey, thinly laminated, organic-rich “shales” (Fig. 2). This formation is characterized by the abundance of ammonite compressions (e.g., Harpoceras falciferum (Sowerby, 1820), Dactylioceras spp.), aptychii, belemnite rostra and large wood trunks. In other sites from the Causses Basin, the “Schistes Cartons” also yielded rare vertebrate remains (e.g., crocodiles, ichthyosaurs; Bomou et al., 2021). At Saint Bauzile, the first meter of the formation displays an alternation of centimetric, orange and oxidized shale beds with centimetric black shale beds. The lower part of the formation shows two hard and thinly laminated limestone beds. The first one, regionally called the “Leptolepis bed” by several authors (e.g., Mattei, 1969; Trümpy, 1983), is very fossiliferous, displays a characteristic bituminous smell and constitutes a benchmark bed observed all over the Causses Basin. The two gladius-bearing coleoid specimens were retrieved from this bed. At Saint Bauzile, the “Leptolepis bed” is quite isopach and 20 cm thick. It shows a strong concentration of fish such as Leptolepis cf. coryphaenoides (Bronn, 1830; see Coquel-Poussy, 2013). Near the Saint Bauzile locality, this bed also yields rare crustaceans (Gabalerion; Audo et al., 2017). In the Causses Basin, the biozone associated with the “Leptolepis bed” is variable depending on the localities and the authors. Based on ammonites, most authors stratigraphically locate these beds in the Serpentinum Zone (e.g., Harazim et al., 2013; Pinard et al., 2014; Gatto et al., 2015), others in the Tenuicostatum Zone (e.g., Fonseca et al., 2018). In the Balduc area, the detailed biostratigraphic analysis conducted by Harazim et al. (2013) demonstrated that the “Leptolepis bed” corresponds to the Serpentinum Zone. Regionally, the Marnes de Fontaneilles Formation consists of grey to blue marls yielding a diversified, pyritized, marine, middle to upper Toarcian fauna mainly including ammonites (e.g., Mattei, 1969; Mattei, Combémorel & Enay, 1987; Jattiot et al., 2015b), belemnites (e.g., Pinard et al., 2014), bivalves (Fürsich et al., 2001), gastropods (e.g., Gatto et al., 2015) and rare vertebrates remains (e.g., Sciau, Crochet & Mattei, 1990; Bomou et al., 2021).

Figure 2 Stratigraphic section of the Saint Bauzile site showing the location of the coleoid-bearing bed (black star = “Leptolepis bed”).

Thi., thickness (m); Form., formations; Lith., lithology.

The Toarcian sediments of the Causses Basin were deposited in a shallow epicontinental sea located at a paleolatitude of 25 to 30°N. At the base of the “Schistes Cartons” Formation, Bomou et al. (2021) documented a negative carbon isotope excursion and higher mercury fluxes that were linked with the Toarcian Oceanic Anoxic Event (T-OAE). This event, originating from the intense volcanic activity of the Karoo Ferrar igneous province, is characterized by a widespread deposition of organic-rich shales concomitant with the onset to an episode of global warming. Bomou et al. (2021) showed that the deposition of the “Schistes Cartons” Formation took place during a prolongated period of widespread oxygen-deficiency and elevated carbon burial.

Results

Among Mesozoic gladius-bearing coleoids, three different morphotypes of gladii can be recognized: prototeuthid, loligosepiid and teudopseid (Fuchs, 2009). Each is associated with the corresponding suborders Prototeuthina Naef, 1921, Loligosepiina (Jeletzky, 1965) and Teudopseina Starobogatov, 1983 (Fig. 3), respectively. The two individuals described in this study belong to the Prototeuthina and Loligosepiina, indicating that at least two of the three Mesozoic gladius-bearing suborders were present in this locality.

Figure 3 Morphology, terminology and measurements of the three different gladius morphotypes among Mesozoic gladius-bearing coleoids.

Data source credit: Marroquín, Martindale & Fuchs (2018).

Subclass Coleoidea Bather, 1888	
Superorder Octobrachia Haeckel, 1866	
Suborder Prototeuthina Naef, 1921	

Diagnosis (after Fuchs, 2020). Octobrachiates with torpedo-shaped body; gladius length (= median field length) equals mantle length; gladius with triangular median field and ventrally closed (funnel-like) conus; gladius very slender to moderately wide, maximum gladius width usually coincides with maximum median field width (by contrast to Loligosepiina and Teudopseina); median field slender (compared to most loligosepiids and teudopseids), with median and lateral reinforcements; lateral reinforcements and central median field may be projected; median field area large to very large compared to lateral fields (gladius is median field dominated); hyperbolar zone indistinct or absent, hyperbolar zone length to median field length <0.6; lateral fields very slender to moderately wide.

Family Plesioteuthidae Naef, 1921	
Type genus. PlesioteuthisWagner, 1859	

Diagnosis (after Fuchs, 2020). Medium-sized prototeuthids; gladius very slender to moderately wide (gladius widthmax to gladius length 0.05–0.25), with triangular median field and ventrally closed (funnel-like) conus; median field very slender to slender (median field widthhypz to hyperbolar zone length <0.35 = opening angle <20°); median field area large to very large (median field area to gladius area 0.70–1.0); lateral fields very slender to moderately wide; hyperbolar zone very short to long (hyperbolar zone length to median field length <0.6); median and lateral reinforcements present on the median field; vestiges of septa and guard unknown, eight arms equipped with uniserial circular suckers, sucker-rings absent; arm length variable; funnel-and nuchal-locking cartilages absent; fins terminal; fin shape variable.

Included genera (after Fuchs, 2020). Plesioteuthis Wagner, 1859; Boreopeltis Engeser & Reitner, 1985; Dorateuthis Woodward, 1883; Eromangateuthis Fuchs, 2019; Nesisoteuthis Doguzhaeva, 2005; Normanoteuthis Breton, Strugnell & Donovan, 2013; Paraplesioteuthis Naef, 1921; Romaniteuthis Fischer & Riou, 1982; Rhomboteuthis Fischer & Riou, 1982; Senefelderiteuthis Engeser & Keupp, 1999.

Stratigraphical and geographical range (after Fuchs & Larson, 2011a). (?)Late Triassic (Rhaetian), Early Jurassic (Toarcian)–Late Cretaceous (Maastrichtian); Europe, Central Russia, Lebanon, North America and Australia.

Genus Paraplesioteuthis Naef, 1921

Type species. Geoteuthis sagittata Münster 1843 by the subsequent designation of Naef (1922, p. 111).

Diagnosis (after Fuchs, 2020). Gladius medium-sized, slender to moderately wide (gladius width max to gladius length 0.15–0.25) with a bipartite median ridge. Median field slender to moderately wide (median field widthhypz to hyperbolar zone length 0.25–0.35 = opening angle 14°–20°), triangular and with lateral platelike reinforcements. Lateral reinforcements and central median field anteriorly projected. Median field area large to very large (median field area to gladius area 0.75–0.85). Lateral fields slender (lateral fields widthmax to median field widthmax 0.85–0.95). Hyperbolar zone moderately long to long (hyperbolar zone length to median field length 0.45–0.55). Soft parts poorly known.

Included species. Paraplesioteuthis sagittata (Münster 1843) only.

Stratigraphical and geographical range. Upper Pliensbachian–lower Toarcian; southern Germany (Holzmaden region; Fuchs, 2006b), western Canada (Fernie Formation; Hall, 1985, Marroquín, Martindale & Fuchs, 2018), southeastern France (Causse Basin; this study).

Paraplesioteuthis sagittata (Münster 1843)	
figs. 4 and 5	
1843. Geoteuthis sagittata Münster, pp. 672–673, pl. 7, fig. 3, pl. 8, fig. 4.	
p 1843. Geoteuthis hastata Münster, p. 73, pl. 14, fig. 4.	
non 1843. Geoteuthis hastata Münster, p. 73, pl. 8, fig. 3.	
1860. Geoteuthis sagittata Münster; Wagner, 1860, p. 807.	
1922. Paraplesioteuthis hastata (Münster); Naef, p. 114, fig. 41a–c.	
1978. Paraplesioteuthis sagittata (Münster); Reitner, p. 210, fig. 6.	
1984Paraplesioteuthis sagittata (Münster); Riegraf et al. p. 36.	
1984. Paraplesioteuthis hastata (Münster); Riegraf et al. p. 36.	
1985. Paraplesioteuthis hastata (Münster); Hall, p. 871, fig. 1.	
1990. Paraplesioteuthis sagittata (Münster); Doyle, p. 205.	
2006b. Paraplesioteuthis hastata (Münster); Fuchs, pl. 16a–c.	
2009. Paraplesioteuthis hastata (Münster); Fuchs, fig 1. a–b.	
2011a. Paraplesioteuthis sagittata (Münster); Fuchs & Larson, fig 7.1.	
? 2018. Paraplesioteuthis cf. sagittata (Münster); Marroquín et al., figs. 6–10.	
2020. Paraplesioteuthis sagittata (Münster); Fuchs, p. 10, fig. 4,1a–b	

Holotype. The original specimen of Münster (1843, pl. 7, fig. 3) was lost during World War II.

Lectotype (designated by Marroquín, Martindale & Fuchs, 2018). Original of Münster (1843, pl. 8, fig. 4), Geologisch-Paläontologisches Museum Tübingen, GPIT 1529-2 (original of Reitner, 1978, fig. 6).

Type locality. Holzmaden region, southern Germany.

Type horizon. Posidonia Shales Formation, lower Toarcian (Lower Jurassic).

Stratigraphical and geographical range. As for genus.

Material. One incomplete specimen (M486_2024.1.1) from the lower Toarcian “Schistes Cartons” Formation of the Causses Basin, in the vicinity of Saint Bauzile village (Lozère, France).

Description. Although the single specimen (M486_2024.1.1) is only partially preserved, morphological description and taxonomic identification at the species level remain possible. The specimen consists of a long and slender gladius (interpreted here as in dorsal view) with a triangular, anteriorly diverging median field (Figs. 4A–4C). The preserved gladius length (= median field length) is 203 mm. Although the anterior end of the gladius is not preserved, we hypothesize that the original gladius length did not exceed 220 mm (according to Klug et al., 2021a, gladii of Paraplesioteuthis sagittata rarely reach 200 mm). Of note, we suspect that the anterior part of the gladius, which is poorly preserved, was affected by slight disruptions and distortions (Figs. 4A, Fig. 4B). In our opinion, this impedes providing a reliable measurement of the anteriormost gladius width. Partially preserved lateral reinforcements (most conspicuous in the posterior part of the gladius, Figs. 4A–4C, 5) diverge from posterior to anterior extremities. Based on the estimated median field width hypzto hyperbolar zone length (see Fig. 3 and Appendix S1A), we estimate that the lateral reinforcements form an opening angle of ∼11° (see Appendix S1A). The lateral fields are relatively slender. Although the outline of the hyperbolar zone is hardly discernible, it appears relatively long (estimated hyperbolar zone length to median field length ratio is 0.44). Finally, it cannot be determined whether the median ridge is bipartite as commonly described for Paraplesioteuthis representatives.

Remarks. In our opinion, the gladius is too poorly preserved to provide an accurate measurement of the original gladius widthmax. Nevertheless, based on the general shape of the gladius, we consider that the original gladius widthmax to gladius length ratio likely fell within the range of values mentioned by Fuchs (2020, p. 10) for Paraplesioteuthis (i.e., 0.15–0.25). Of note, the opening angle of ∼11°  framed by the lateral reinforcements in the present specimen is lower than the range of values given by Fuchs (2020, p. 10) for Paraplesioteuthis (i.e., 14°−20°). On the other hand, it is comparable to the opening angle of 10°  mentioned by Hall (1985) for the P. hastata (Münster 1843) specimen from western Canada (P. hastata is regarded as conspecific with P. sagittata by Fuchs & Larson, 2011a). Thus, we suggest that values of opening angle for Paraplesioteuthis should be redefined as ranging from about 10° to 20°.

According to Fuchs & Larson (2011b), the Middle Jurassic genus Romaniteuthis differs from Paraplesioteuthis by having a reduced median field width (i.e., gladius widthmax to gladius length ratio 0.05–0.15; Fuchs, 2020) and lateral reinforcements as keels. Although the specimen described herein is too poorly preserved to provide a reliable measurement of the preserved gladius widthmax, its original gladius widthmax to gladius length ratio was probably not less than 0.15. Furthermore, it does not exhibit prominent lateral keels.

Paraplesioteuthis lateral fields are relatively short (Fuchs & Larson, 2011a; see Fig. 4D, Fig. 4E). In this regard, the lateral fields of the present specimen seem more similar to that of Romaniteuthis, since they appear slightly elongated, in oval shape (compare Figs. 4A–4C, 5 with fig 7.2 in Fuchs & Larson, 2011a). This may however be due to slight intraspecific variability. Finally, the estimated hyperbolar zone length to median field length ratio for the present specimen is 0.44, which nearly falls within the range of values given by Fuchs (2020) for Paraplesioteuthis (i.e., 0.45–0.55).

In sum, despite a possible slight difference in shape of lateral fields, we consider that other features of the present gladius support its attribution to the species Paraplesioteuthis sagittata.

Order Vampyromorpha Robson, 1929

Suborder Loligosepiina Jeletzky, 1965

Diagnosis (after Fuchs, 2020). Small- to large-sized octobrachiates with bullet-shaped body; gladius length (= median field length) equals mantle length; gladius with triangular median field and cup-shaped conus; gladius slender to wide, maximum gladius width always exceeds maximum median field width; median field width very slender to moderately wide without pronounced median reinforcements, anterior median field margin concave, straight or convex; median field area small to large; hyperbolar zone mostly well-arcuated, rarely indistinct, long to very long; lateral fields usually moderately wide.

Family Loligosepiidae Regteren Altena, 1949

Type genus. Loligosepia Quenstedt, 1839.

Diagnosis (after Fuchs, 2020). Medium-sized loligosepiids; gladius slender to wide (gladius widthmax to gladius length 0.10–0.60), with deeply concave (V-shaped) hyperbolar zone; median field very slender to moderately wide (median field widthhypz to hyperbolar zone length 0.10–0.40 = opening angle 7°–23°), anterior median field margin slightly convex; median field area small to moderate (median field area to gladius area 0.35–0.45); hyperbolar zone very long (hyperbolar zone length to median field length 0.85–0.95); lateral fields moderately wide (lateral fields widthmax to median field widthmax 1.30–1.85), anterior limit of lateral fields clearly pointed (spine-like); inner and outer asymptotes ridge-like.

Included genera. Loligosepia Quenstedt, 1839 and Jeletzkyteuthis Doyle, 1990.

Stratigraphical and geographical range. Lower Sinemurian–lower Toarcian; Germany, Luxembourg, France, Switzerland, UK, Canada (Alberta).

Genus Loligosepia Quenstedt, 1839

Type species. Loligo aalensis Schübler in Zieten, 1832 (by subsequent designation of Regteren Altena, 1949, p. 57) from the lower Toarcian Posidonia Shales of Holzmaden (Germany).

Diagnosis (after Fuchs, 2020). Medium-sized loligosepiids, gladius moderately wide to wide (gladius widthmax to median field length 0.30–0.60); median field slender to moderately wide (median field widthhypz to hyperbolar zone length 0.20–0.40 = opening angle 12°−23°); anterior median field margin convex; median field area small to moderate (median field area to gladius area 0.35–0.45); hyperbolar zone very long (hyperbolar zone length to median field length 0.85–0.95); lateral fields moderately wide (lateral fields widthmax to median field widthmax 1.35–1.80; arms short to moderate (arm length to mantle length ∼0.45).

Included species. Loligosepia bucklandi (Voltz, 1840) from the lower Sinemurian of Dorset (UK) and L. aalensis (Schübler in Zieten, 1832) from the lower Toarcian of Holzmaden.

Stratigraphical and geographical range. Lower Sinemurian–lower Toarcian of Germany, Luxembourg, France, UK and Canada (Alberta).

?Loligosepia sp. indet.

Figures 6 and 7

Figure 4 Paraplesioteuthis sagittata (Münster, 1843), specimen M486_2024.1.1 (lower Toarcian, Saint Bauzile, Causses Basin) in dorsal view, under natural (A) and UV (B) light. (C) Interpretative drawing of specimen M486_2024.1.1. (D) Paraplesioteuthis sagittata specimen from the lower Toarcian Posidonia Shale Formation, Germany (UMH collection, (Fuchs, 2020), Fig. 4-1a Dirk Fuchs. 2020. Part M, Chapter 23G: Systematic Descriptions: Octobrachia. Treatise Online 138: p. 12). (E) Gladius reconstruction of Paraplesioteuthis sagittata (Fuchs, 2020, Fig. 4-1b).

Scale bars: 10 mm.

Figure 5 Paraplesioteuthis sagittata (Münster, 1843), close-up of the posterior part of specimen M486_2024.1.1 (lower Toarcian, Saint Bauzile, Causses Basin) under natural light.

Scale bar: 10 mm.

Figure 6 ?Loligosepia sp. indet., specimen M486_2024.1.2 (lower Toarcian, Saint Bauzile, Causses Basin) in dorsal view, under natural (A) and UV (B) light. (C) Interpretative drawing of specimen M486_2024.1.2.

Scale bar: 10 mm.

Figure 7 ?Loligosepia sp. indet., close-up of the posterior part of specimen M486_2024.1.2 (lower Toarcian, Saint Bauzile, Causses Basin) under natural light.

Scale bar: 10 mm.

Material. One incomplete specimen (M486_2024.1.2) from the lower Toarcian “Schistes Cartons” Formation of the Causses Basin, in the vicinity of Saint Bauzile village (Lozère, France).

Description. The preserved gladius length (= median field length) of this specimen (M486_2024.1.2) is 76 mm and the maximum preserved gladius width is approximatively 33 mm. Although a significant part of the original gladius is most certainly missing, we tentatively hypothesize that the original gladius did not exceed 150 mm. Unfortunately, few features can be described on this specimen. On the posterior part of the specimen, inconspicuous lines are interpreted as a disrupted median line and inner asymptotes (Figs. 6A–6C, 7). The black structure preserved anteriorly is interpreted as the ink sac (Figs. 6A–6C). The specimen outline appears weakly constricted posteriorly, although it cannot be determined whether it is a genuine feature, or if it is due to slight taphonomic disruptions. The median field widthmax of the original gladius cannot be assessed.

Remarks. The overall shape of the gladius combined with the presence of lines interpreted as a median line and inner asymptotes hint at the possibility that this specimen belongs to the suborder Loligosepiina. Mesozoic gladius-bearing coleoids belonging to the suborder Teudopseina exhibit a characteristically constricted median field anteriorly, which does not seem to be the case in specimen M486_2024.1.2. Within the suborder Loligosepiina, the assignment of this specimen to the genus Loligosepia is unsettled. It is only based on a conjectural original size of the specimen (estimated around 150 mm) that is comparable with that of representatives of the Loligosepia species L. bucklandi (Voltz, 1840) and L. aaalensis (Schübler in Zieten, 1832). In our opinion, Jeletzkyteuthis species differ mostly by their larger gladius size. For example, J. coriaceus (Quenstedt, 1849) gladiuses regularly exceed 200 mm in length, according to Klug et al. (2021a). Besides, the poor preservation of both specimens prevents a robust comparison between the present specimen and the J. coriaceus specimen from the coeval locality of Langres (Guérin-Franiatte & Gouspy, 1993). Ultimately, the hypothesis that the present specimen would represent a juvenile of a Jeletzkyteuthis cannot be discarded.

Discussion: Paleobiogeography of Paraplesioteuthis sagittata

The present record of Paraplesioteuthis sagittata is only the second one in Europe and the third in the world (western Canada, Germany and now France; Fig. 8). During the Early Jurassic, the NW Tethyan and Artic realms (which consisted of two epicontinental seas) were linked by a narrow seaway named the Viking Corridor (Ziegler, 1988; Fig. 8). In this time interval, the Serpentinum Zone (from where the gladius-bearing coleoids described herein come) marks the onset of the disruption of a previous ammonite provincialism, with a strong homogenization of all Tethyan and Arctic ammonite species (Dera et al., 2011). This event is linked with the origination of numerous cosmopolitan ammonite taxa in the Mediterranean domain (Macchioni & Cecca, 2002). In this context, the Viking Corridor probably regulated the mixing between Artic and Euro-Boreal ammonites (Smith, Tipper & Ham, 2001; Dera et al., 2011).

Figure 8 Worldwide occurrences of Paraplesioteuthis sagittata.

Worldwide occurrences of Paraplesioteuthis sagittata (black stars, 1 = western Canada, 2 = southern Germany and 3 = southeastern France) in the paleogeographical context of the Pliensbachian −Toarcian interval. (A) Global paleogeography. (B) Paleogeographical details of the NW Tethyan realm (scale bar = 250 km). Maps are modified after Dera et al. (2010).

Based on this, it can be hypothesized that some Early Jurassic gladius-bearing coleoids (such as P. sagittata), similarly to ammonites, broadly originated in the NW Tethyan realm and moved to the Arctic realm through the Viking Corridor, to eventually move even farther to North America (in the context of the Early Toarcian global warming event; see, e.g., Dera & Donnadieu, 2012). Other routes cannot be excluded, such as the Hispanic Corridor (Fig. 8), a narrow epicontinental seaway that was sporadically active since the Late Sinemurian −Early Pliensbachian time interval (Aberhan, 2001; Aberhan, 2002; Venturi, Bilotta & Ricci, 2006; Dera et al., 2009). However, according to Dera et al. (2011, p. 100), the Hispanic Corridor “…was certainly too shallow for allowing massive movements of hemipelagic organisms such as ammonites”. Since gladius-bearing coleoids are also hemipelagic organisms, we presume that gladius-bearing coleoids, similarly to ammonites, were not able to go through the Hispanic corridor.

In the hypothesis that gladius-bearing coleoids moved from the Mediterranean domain to North America through the Viking Corridor, we would expect to find P. sagittata specimens in localities from the Artic Realm, provided that the geological time interval is represented and that there are geological horizons peculiarly favourable to the preservation of gladius-bearing coleoids.

Conclusions

Two of the three suborders of Mesozoic gladius-bearing coleoids are present in the studied material, which comprises so far only two specimens. This implies a previously unrecognized early Toarcian gladius-bearing coleoid diversity in the Causses Basin and points out the need for further field investigations in the lower Toarcian black shales in this area. Finally, based on the known worldwide occurrences of P. sagittata, we tentatively suggest that this species originated in the Mediterranean domain and moved to the Arctic realm through the Viking Corridor to eventually move even farther to North America.

Supplemental Information

Supplemental Information 1 Specimen measurements

The authors are indebted to Christian Klug, Dirk Fuchs and an anonymous reviewer whose suggestions greatly improved the manuscript. We also thank P. Loubry and L. Cazes for photographing the specimens, as well as A. Lethiers for his work on the figures.

Additional Information and Declarations

Competing Interests

Author Contributions

Data Availability

The authors declare there are no competing interests.

Romain Jattiot conceived and designed the experiments, performed the experiments, analyzed the data, prepared figures and/or tables, authored or reviewed drafts of the article, and approved the final draft.

Nathalie Coquel-Poussy conceived and designed the experiments, analyzed the data, authored or reviewed drafts of the article, and approved the final draft.

Isabelle Kruta conceived and designed the experiments, performed the experiments, analyzed the data, prepared figures and/or tables, authored or reviewed drafts of the article, and approved the final draft.

Isabelle Rouget conceived and designed the experiments, performed the experiments, analyzed the data, prepared figures and/or tables, authored or reviewed drafts of the article, and approved the final draft.

Alison J. Rowe conceived and designed the experiments, performed the experiments, analyzed the data, prepared figures and/or tables, authored or reviewed drafts of the article, and approved the final draft.

Jean-David Moreau conceived and designed the experiments, analyzed the data, prepared figures and/or tables, authored or reviewed drafts of the article, and approved the final draft.

The following information was supplied regarding data availability:

The specimen measurements are available in the Supplemental File.

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
