# Peer review of "The first gladius-bearing coleoid cephalopods from the lower Toarcian “Schistes Cartons” Formation of the Causses Basin (southeastern France)"

_PeerJ, doi:10.7717/peerj.16894_

## Round 0.1 · original submission · Minor Revisions

Dear Authors,

Thank you very much for submitting your research to PeerJ.

I am glad to say that after minor revision, your manuscript will be accepted. Please, take into account all reviewers' comments, including those directly in the text file (see attachments).

·

Basic reporting

Dear authors,
this is a nice, concise, short, and interesting contribution on Toarcian coleoids from France.
It is well-written, the literature is well-covered, the structure is straight forward.

Experimental design

The report follows the classic standards of such descriptive articles.
The research question could be somewhat better sold.

Validity of the findings

It surprises me that no other coleoids have been described from the Schistes cartons before, because I vaguely remember having seen material from France.
In any case, It is very welcome that this material is properly described.
I have a few remarks/ questions:
1. Why do your write that these findings are unexpected? The facies is reasonably similar to the coeval strata from Germany, which isn't too far away. I found it more surprising that they weren't described earlier. Honestly, I suspect a strong collection bias there since in Germany, these strata are mined in numerous pits since centuries. Hence, either you remove the word 'unexpected' or you explain it.
2. The most interesting aspect of the whole paper is the palaeogeographic distribution. How did the coleoids get around northern America into Panthalassa? How was the cllimate, wasn't it too icy in the Viking corridor at the time? At least in the Pliensbachian of Germany, there are already glendonites.
3. There is a further possibility: They could have migrated along the Tethys and then crossed Panthalassa. Or can you rule this out? I think these animals were excellent swimmers.
4. Romain, you know Triassic ammonoids quite well. I remember there were shared occurrences of cephalopods in the Tethys AND in eastern Panthalassa. This might be worth mentioning and discussing.

Additional comments

Above all, this is a nice short study.
I think the aims should be formulated a bit better, the discussion could be given a bit more weight, and then this will be a nice contribution for PeerJ.
Best regards,
Christian (Klug)

·

Basic reporting

The literature references are in some parts too extensive, i.e. may be shorter.
Figures 4+5 may additionally need close-ups in order to better comprehend the morphologic description of the specimens.

Experimental design

no comment

Validity of the findings

no comment

Additional comments

I greatly appreciate the introduction of a new coleoid locality that surprisingly challenges the most famous Konservat-Lagerstaetten world wide.

Reviewer 3 ·

Basic reporting

I suggest to consider previous reports of gladius-bearing coleoids from the study area. These can be found for instance in publications by J. Sciau (see the pdf attachment for more details). This additional information has some potential outcome for the conclusions that the authors draw.

Otherwise, the article is complete, well-written with sufficient field background.

Experimental design

no comment

Validity of the findings

The findings are valid

Additional comments

Please find some minor corrections and suggestions in the attached pdf document

Annotated reviews are not available for download in order to protect the identity of reviewers who chose to remain anonymous.

---

## Round 0.2 · accepted · Accept

Thank you for addressing all reviewers' comments, which I find satisfactory and exhaustive. The manuscript is ready for publication.